# Dental Caries in Children Under Five Years of Age in Mongolia

**DOI:** 10.3390/ijerph17134741

**Published:** 2020-07-01

**Authors:** Mizuho Nishino, Bazar Amarsaikhan, Nanayo Furumoto, Saki Hirao, Hiroko Bando, Akemi Nakagawa, Sukhbaatar Nomingerel, Begzsuren Bolorchimeg, Masami Fujimoto

**Affiliations:** 1Tokushima Mongol Medical Exchange Association, Tokushima-shi, Tokushima 779-3111, Japan; 2Vice President for Research and Development, Mongolian National University of Medical Sciences, Ulaanbaatar 14210, Mongolia; amarsaikhan@mnums.edu.mn; 3Faculty of Human Life Sciences, Tokushim a Bunri University, Tokushima-shi, Tokushima 770-8514, Japan; furumoto@tks.bunri-u.ac.jp (N.F.); m-fujimoto@tks.bunri-u.ac.jp (M.F.); 4Tominaga Dental Clinic, Naruto-shi, Tokushima 771-0360, Japan; hirao@emat.jp; 5Hayakumo Dental Clinic, Tokushima-shi, Tokushima 770-0006, Japan; sora.1192@icloud.com; 6Medical Corporation Kenshokai, Tokushima-shi, Tokushima 779-310, Japan; d.h.nakagawa@water.ocn.ne.jp; 7Mydent Dental Clinic, Ulaanbaatar 210648, Mongolia; s_nomin77@yahoo.com; 8Mondent Dental Clinic, Ulaanbaatar 210648, Mongolia; bolorchimeg.99@yahoo.com

**Keywords:** caries prevalence, deft, early childhood caries (ECC), Mongolia

## Abstract

During the Japan International Cooperation Agency (JICA) partnership program in Mongolia, six times between October 2017 and October 2019, education for caries prevention, a questionnaire of daily oral health behavior, and an oral examination for parents and children aged 0–5 years old were done. The target parent population were middle socio-economic class families. In total, there were 2223 participants. The caries prevalence measured in October 2017, January, April, and October 2018, and April and October 2019, was 78.0% (95% CI: 74.2–81.4); 79.4% (73.7–84.4); 80.8% (76.2–84.9); 76.4% (70.1–82.0); 89.3% (85.3–92.6); and 82.6% (79.3–85.6), respectively. Compared to October 2017, in October 2019, the deft of three years old was significantly decreased (*p* < 0.01) and that of four years old was also decreased (*p* = 0.085). For the prevention of early childhood caries (ECC), daily oral health behaviors are important. In 2019, compared with the percentage of 0–5 years old in 2017, the frequency of tooth cleaning per day, parents cleaning after children, and parents watching during children’s tooth cleaning were significantly increased (*p* < 0.01). Unfortunately, the frequency of sugary—snack intake per day showed an increased tendency (*p* < 0.05). The baby teeth decayed, extracted and filled (deft) index at three and/or four years old in October 2019 was correlated with the childrens’ background characteristics, such as parent educational attainment, frequency of sweets intake, frequency of daily tooth cleaning, and parental cleaning of children’s teeth. The effects of the JICA program were recognized.

## 1. Introduction 

Dental caries is highly prevalent globally and a significant public health problem, as it reduces quality of life [1,2]. The majority of caries lesions are concentrated in a few, often disadvantaged-social groups. A systematic review of socioeconomic inequality and caries showed that a low socioeconomic position is associated with a higher risk of having caries. The association between lower educational background and having decayed, missing, and filled teeth (DMFT)/deft > 0 was significantly increased in highly developed countries [3]. The association between breastfeeding and dental caries is complicated. Breastfeeding during the day beyond the age of 12 months is not associated with caries, but infants who were breastfed at night > 2 times were associated with caries [4]. Most children with rampant or nursing caries had an unbalanced diet with a high sugar content. Breastfed infants younger than 6 months or older than 12 months showed rampant caries. Their meals were unusually sweetened in infancy [5].

Early childhood caries (ECC) is defined as the presence of one or more decayed (non-cavitated lesions), missing (due to caries), or filled tooth surfaces in any primary tooth in a preschool-aged child between birth and 71 months of age [6].

ECC has a significant influence on individuals, families, and societies. The disease affects primary and permanent teeth, and influences the general health and quality of life across the entire course of one’s life. ECC is linked with other frequent diseases of childhood, primarily due to risk factors shared with other noncommunicable diseases (NCDs) such as high sugar intake, and diseases related to other health conditions such as obesity. Severe dental caries is associated with poor growth [7]. The World Federation and World Health Organization (WHO) indicated that more than 200 diseases are caused by dental caries [8].

The government of Mongolia approved their second “National Oral Health” program in 2006. This was a development with the expectation of a reduction of caries prevalence. Despite the national program, the dental caries prevalence rate in Mongolia is still very high [9,10,11].

Fortunately, ECC is preventable, with almost all risk factors modifiable. ECC differs from dental caries in older children and adults in its rapid development, its diversity of risk factors, and in the control of disease. ECC is influenced strongly by the health behaviors and practices of children, families, and caregivers [7].

This study aimed to prevent ECC in Mongolia through the technology transfer of an oral health examination system and tooth cleaning methods by parents, and through the improvement of health behaviors and practices of children, families, and caregivers in kindergartens.

## 2. Materials and Methods

For caries prevention in children in Mongolia, the Japan International Cooperation Agency (JICA) Partnership Program “Prevention of Dental Caries in Infants in Mongolia” was run from 2017 to 2019 in accordance to the Mongolian JICA Office, the Tokushima Mongol Medical Exchange Association, the Mongolian Dental Association, the Deputy Governor of the Capital City of Mongolia, and the Development Financing Department, Ministry of Finance, Mongolia.

The target area was an urban area, in Mongolia: 18 Khoroo, Bayansurkh district, Ulaanbaatar. Target Groups were dentists, parents, and children. The chief of the District Health Center invited 0–1 year old (0–23 months) children to the District Health Center. The chief of public kindergarten no. 129 informed parents of 2–5 years old (24–71 months) children.

Informed consent for the education to parents/guardians, questionnaire survey, and dental examination was obtained from all the parents/guardians.

Mongolian dentists trained twice for the oral health examination system, i.e., parental education, and taking the questionnaire on daily oral health behavior and oral examination, in Tokushima, Japan.

Activities in the target area were done six times (Table 1) by two Japanese dentists (MN and SH), two Japanese dental hygienists, and six Mongolian dentists who worked together. The Mongolian dentists educated the parents with slides and an education sheet (Figure 1).

The “Assessment by Parents/Guardians for Oral Health and Risks of Children in Mongolia based on World Health Organization questionnaire” was taken. The questionnaire consisted of information on a child’s background, such as sex, age, frequency of daily tooth cleaning, frequency of daily sugary foods and drinks, complete tooth cleaning by parent after child brushing, dental office visit, and parental educational attainment.

The knee-to-knee oral examination with the “Oral Health Assessment Form for Children in Mongolia based on World Health Organization” were performed by two Japanese dentists (MN and SH) using a TePe Mini ^TM^ Soft toothbrush (TePe, Malmo, Sweden), a DENT. Clip Mirror with DENT. Clip Light II (LION Dental Products Co., Tokyo, Japan), and a (WHO)YDM explorer 01-221 (YDM Corporation, Tokyo, Japan) dental probe.

The diagnostic criteria for caries followed the WHO recommendations [12]. Two Japanese examiners (MN and SH) were the accredited specialists for pediatric dentistry having many experiences of oral examination. Inter-reliability between the two examiners and intra-reliability of examinations were high.

Caries experience was measured with caries prevalence rate and the deft index. After the dental examination, a child’s oral health report was given to their parents by a Mongolian dentist.

A toothbrush that was used before the oral examination for the removal of dental plaque and the tooth cleaning calendar for encouraging tooth cleaning (Figure 2) were given to each child.

This study used a repeated cross-sectional observational study to examine the effects of this project.

No dental treatment was provided on site. The results of the oral examination and questionnaire surveys were analyzed using the IBM SPSS Statistics Ver24 (IBM, New York, NY, USA, ). Data were evaluated by using Student’s *t*-test or the chi-squared test.

## 3. Results

This study included a total of 2223 participants, comprising 1051 (47.3%) boys and 1172 (52.7%) girls (Table 1). Caries prevalence rates are shown in Figure 3.

Caries prevalence in October 2017, January, April, and October 2018, and April and October 2019, was 78.0% (95% CI: 74.2–81.4); 79.4% (73.7–84.4); 80.8% (76.2–84.9); 76.4% (70.1–82.0); 89.3% (85.3–92.6); and 82.6% (79.3–85.6), respectively. 

The deft in October 2017 and 2019 is shown in Figure 4. At age 3, the deft of 2019 was significantly lower than that of 2017 (*p* < 0.01), at age 4, the deft of 2019 had a lower tendency than that of 2017 (*p* = 0.085).

Questionnaire answers for October 2017 and 2919 are shown in Table 2.

Frequency of tooth cleaning per day, parental tooth cleaning after the child itself, and a parent watching the child during the child is tooth cleaning were significantly increased (*p* < 0.01). Unfortunately, the frequency of sugary-snack intake per day had an increased tendency (*p* < 0.05).

Parental educational background, sex, frequency of sweets intake per day, frequency of daily tooth cleaning, parental cleaning of child’s teeth affected the deft at 3 and/or 4 years of age (Figure 5).

## 4. Discussion

In this study, the participants were all children who came in for an oral examination. Researchers planned a follow-up for all children, but in Mongolia, almost all parents work, and it was difficult to come every time for an oral examination. The definition of the project target has not changed, but its contents have changed. This study was a repeated cross-sectional observational study to examine the effect of this project.

This study aimed at the technology transfer of an oral examination system in early childhood and tooth cleaning methods by parents. These purposes were achieved. Japanese dentists educated Mongolian dentists and the parents/guardians of children on how to brush an infant’s teeth, as illustrated in Figure 1 (educational sheet). Another purpose was the improvement of oral health behavior and practices of children and families. The results of this study showed that caries prevalence in Mongolian children is still high. However, the deft of three years old in October 2019 was significantly lower than that of October 2017 (*p* < 0.01). This project started in October 2017. Parents of three years old children in 2019, were educated since their child was 0–1 years old. In this group, a purpose of this study, the improvement of oral health behavior might have been achieved.

Unfortunately, the restriction of sugary foods and drinks intake is very difficult. Effective health education is necessary. The socioeconomic status of this study’s recipients was middle-class families. With higher parental educational attainment at three years old in 2019, the deft was lower than that in 2017. Global evidence suggests that higher educational attainment is associated with better health outcomes.

A report of the actual condition of oral diseases in Japan, 2016, showed that the caries prevalence and the deft index of five years old were 39.0% and 1.7, respectively [13]. In this research, in Ulaanbaatar, overall caries prevalence was slightly higher from 78.0% (2017) to 82.6% (2019). However, comparing 2017 and 2019, the proportion of deft > 4 = decreased as follows, 61.5% to 46.0% for three years old, 84.1% to 60.0% for four years old, and 84.2% to 63.6% for five years old.

Over 67% of 3–6 years-old children in Beijing, China experienced caries, a level comparable to other reports from China and other developing countries, but 50% greater than those seen in United States preschool children [14].

Among Southeast Asian countries, caries prevalence of five- to six-years old children ranged from 25% (Myanmar) to 95% (Vietnam), and mean deft score ranged from 0.9 (Myanmar) to 9.0 (Cambodia) [15]. In this study, the caries prevalence of 0–5 years old children in October, 2019 was 82.6%, which is still high.

Breastfeeding during the day beyond the age of 12 months was not associated with caries, but infant who were breastfed at night more than twice were associated with caries [4]. World Health Organization recommends exclusively breastfeeding for the first six months of life, and the introduction of nutritionally-adequate and safe complementary (solid) foods at six months together with continued breastfeeding up to two years of age or beyond [16]. In this study, at the first time of oral examination, in October 2017, a lot of three years old children breast-fed at the oral examination room. During the second visit, a Japanese dentist explained to the parents to complete breastfeeding by 23 months of age.

After the inspection of the Minister of Health, Mongolia of this JICA activity on 9 January 2018, the National Program for Healthy Tooth—Healthy Child was approved during a Cabinet meeting on August 15. With a funding of MNT37 billion, the program was implemented for 2019–2023 [17].

The program aims at reducing dental caries by 30%, and the mean number of decayed, missing and filled teeth (dmft) by 4 points by means of improving the sufficiency and quality of children’s oral health service, ensuring financial stability and expanding inter-sector cooperation. A national program was also implemented providing free dental check-ups and treatment for Mongolian children aged between 2 and 12. In total 336,000 children aged between 2 and 6 received free dental treatment in 2019. For the sustainable renewal of this national program, the distinct effect of reducing dental caries should be proven by 2023.

## 5. Conclusions

In an urban area in Mongolia, six times between October 2017 and 2019, education for caries prevention, a questionnaire for daily oral health behavior, and an oral examination for parents and children aged 0–5 years old were done. Target population of parents were middle socio-economic families. Total participants were 2223, and a repeated cross-sectional observational study was done.

Through the repetition of education, taking questionnaires of daily oral health behavior, and oral examinations, caries prevalence and deft were significantly decreased. However, the restriction of sugary-foods and drinks intake is very difficult. Effective, psychological educational methods are necessary.

## Figures and Tables

**Figure 1 ijerph-17-04741-f001:**
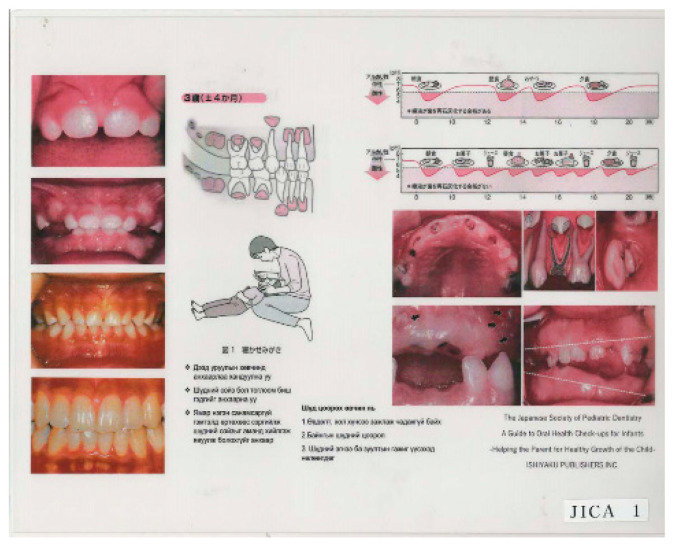
Educational sheet.

**Figure 2 ijerph-17-04741-f002:**
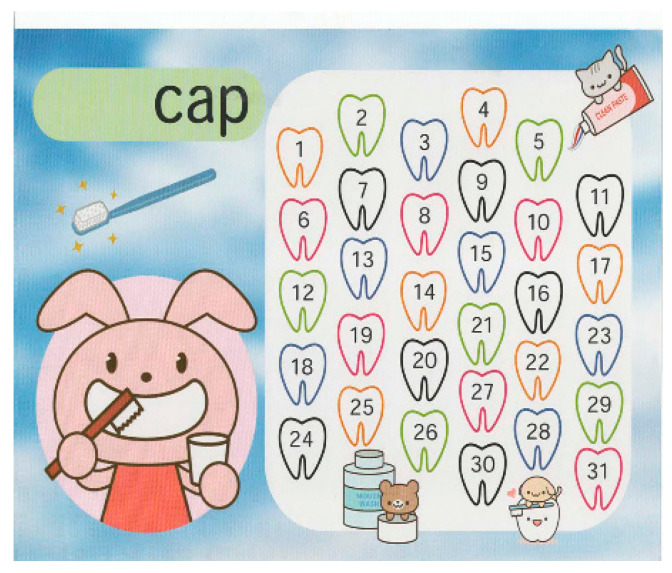
Tooth cleaning calendar.

**Figure 3 ijerph-17-04741-f003:**
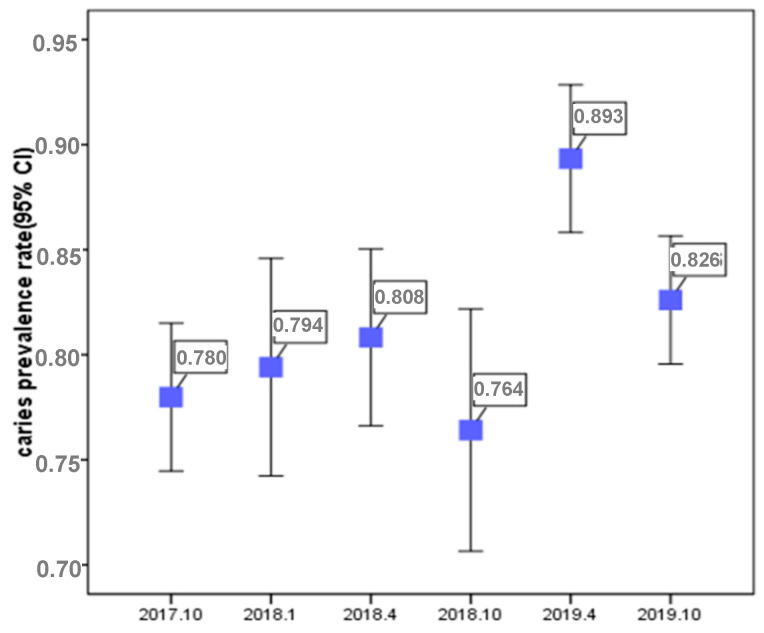
Caries prevalence rate.

**Figure 4 ijerph-17-04741-f004:**
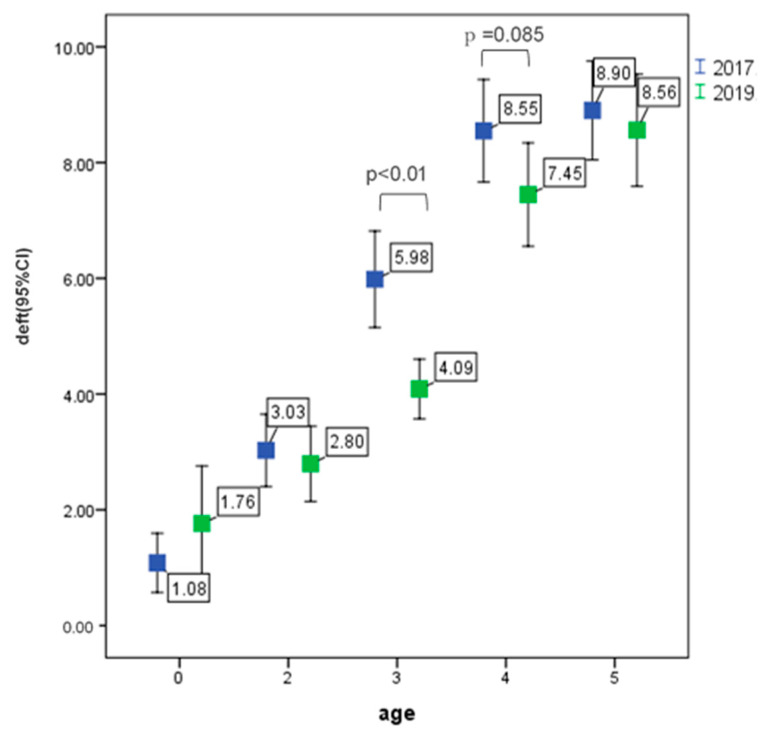
Deft in October 2017 and 2019.

**Figure 5 ijerph-17-04741-f005:**
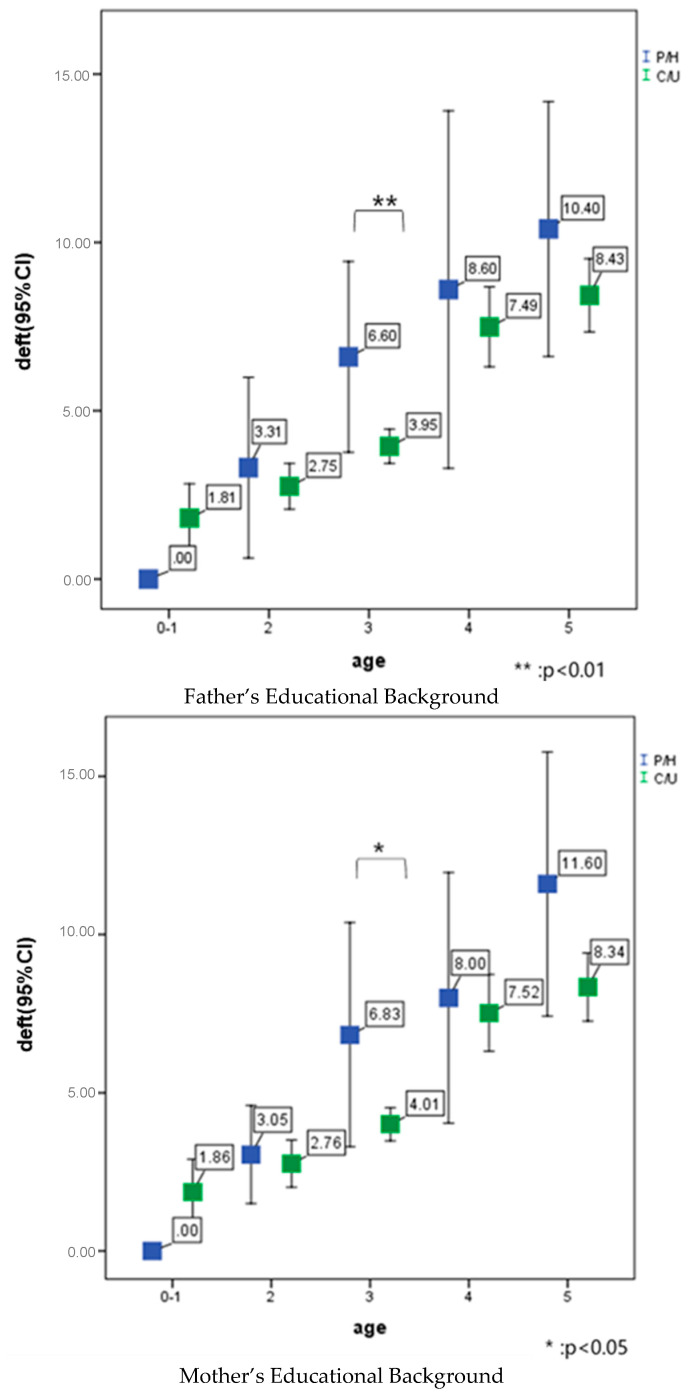
Correlation between parental educational attainment, child sex, frequency of sugary-snack intake per day, frequency of tooth cleaning per day, parental complete tooth cleaning after child itself, and deft index in October 2019.

**Table 1 ijerph-17-04741-t001:** Number of examined children.

Date	0–1 Years Old	2 Years Old	3 Years Old	4 Years Old	5 Years Old	Total	Boys (%)	Girls (%)
October 2017	61	107	122	113	133	536	46.5%	53.5%
January 2018	18	52	48	48	72	238	50.0	50.0
April 2018	17	68	89	86	79	339	45.4	54.6
October 2018	10	36	64	58	44	212	46.2	53.8
April 2019	0	22	77	77	124	300	46.5	53.5
October 2019	38	118	248	105	89	598	48.7	51.3

**Table 2 ijerph-17-04741-t002:** Questionnaire answers.

	October 2017	October 2019	χ^2^ Test
	0–5 Years Old	0–5 Years Old	
Does your child have dental caries?	Number of respondents: 524	Number of respondents: 543	
Yes	62.80%	49.50%	*p* < 0.01
No	37.2	50.5	
Has your child experienced tooth-related pain/discomfort?	Number of respondents: 531	Number of respondents: 540	
Yes	40.90%	12.20%	*p* < 0.01
No	59.1	87.8	
Has your child experience not being able to chew/go to kindergarten because of tooth pain/discomfort?	Number of respondents: 532	Number of respondents: 540	
Yes	18.60%	6.10%	*p* < 0.01
No	81.4	93.9	
Has your child experienced visiting a dental office?	Number of respondents: 529	Number of respondents: 543	
Yes	59.20%	59.70%	NS
No	40.8	40.3	
Reasons for visiting dental office.	Number of respondents: 483	Number of respondents: 442	
Pain/swelling/unable to chew	33.60%	21.70%	*p* < 0.01
Others	66.4	78.3	
Frequency of tooth cleaning per day.	Number of respondents: 527	Number of respondents: 538	
Twice or more	52.10%	63.40%	*p* < 0.01
zero or one	47.9	36.6	
After tooth cleaning by the child itself, does the parent completely clean the child’s teeth again?	Number of respondents: 524	Number of respondents: 519	
Yes	46.60%	68.20%	*p* < 0.01
No	53.4	31.8	
Does your child uses toothpaste?	Number of R respondents: 533	Number of respondents: 542	
Yes	87.60%	90.00%	NS
No	12.4	10	
Does the toothpaste that your child uses contains fluoride?	Number of respondents: 526	Number of respondents: 539	
Yes	59.70%	62.00%	NS
No	40.3	38	
During your child is tooth cleaning, does a parent watch the child?	Number of respondents: 528	Number of respondents: 540	
Yes	71.00%	89.30%	*p* < 0.01
No	29	10.7	
Frequency of sweets intake per day.	Number of respondents: 517	Number of respondents: 539	
Three or more	30.60%	36.90%	*p* < 0.05
Two or less	69.4	63.1	

Child experience of tooth-related pain/discomfort was significantly down (*p* < 0.01).

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
