# Peer review of "Dental Caries in Children Under Five Years of Age in Mongolia"

_ijerph, 2020, doi:10.3390/ijerph17134741_

Round 1

Reviewer 1 Report

This study aimed to prevent ECC in Mongolia through the technology transfer of oral health examination system and tooth cleaning method by parents.

Several issues should be addressed as follows:

  1. The authors did the community projects with the intent to improve the child oral health in Mongolia. The oral health data and questionnaires were collected at different time points. The study design is not clearly explained, e.g. no eligibility criteria of participants, and the sources and methods of selection of participants, sample size calculation, sampling method.
  2. There is a lack of information regarding the numbers potentially eligible, examined for eligibility, included in the study, completing follow-up, and analysed.
  3. Several bias could occur including participation bias, detection bias and selection bias.
  4. Calibration, inter-reliability among several examiners and intra-reliability of examination were not stated.
  5. Several confournding factors were not analyzed and study limitations are not addressed in the discussion.
  6. Due to the cross-sectional study at different time points, the population of each examination were different (not the same person). As seen from the fluctuation of number in each time point, it is unknown whether the differences in def at age 3 and 5 were due to the sampling bias. As seen from the overall caries prevalence was slightly higher from 78% to 83%. The conclusion was not scientifically sound based on the results presented in this format.
  7. Several grammatical errors were detected. Substantive edtiing is required before publication.

Author Response

Ponit 1: The authors did the community projects with the intent to improve the child oral health in Mongolia. The oral health data and questionnaires were collected at different time points. The study design is not clearly explained, e.g.no eligibillity criteria of participants, and the sources and methods of selection of participants, sample size calculation, sampling method.

Response 1: line 58-60 changed

 Target area was urban area in Mongolia: 18 Khoroo, Bayansurkh district, Ulaanbaatar city. Target groups were the dentists, the parents and the children. The chief of the District Health Center called 0-1 year old (0-23 months) children to come to the District Health Center. The chief of public No.129 kindergarten informed to parents of 2-5 years old (24-71 months) children.

Point 2: There is a lack of information regarding the numbers potentially eligible, examined for eligibility, included in the study, completing follow-up, and analysed.

Response 2: line 140 addition

In this study, the number of children was all children who came for oral examinatiointran. Researcher planed follow-up of all children, but in Mongolia almost all parents work and it is difficult to come every time for oral examination. Definition of the project target has not changed, but its contents have changed and each individual contains different people. This study was repeated cross sectional observational study to examine the effect of this project.

Point 3: Several bias could occur including participation bias, detection bias and selection bias.

Response 3:  line 140: as same as Response 2.

Point 4: Calibration, inter-reliability among several examiners and intra-reability of examination were not stated.

Response 4: addition in line 81

   Two Japanese examiners (MN and SH) were the accredited specialists for pediatric dentistry having many experiences of oral examination. Inter-reliability among two examiners and intra-reliability of examinations were high.

Point 5: Several confournding factors were not analyzed and study limitation are not addressed in the discussion.

Response 5: line 140: as same as Response 2.

Point 6: Due to the cross-sectional study at different time points, the population of each examination were different (not the same person). As seen from the fluctuation of number in each time point, it is unknown whether the differences in def at age 3 and 5 were due to the sampling bias. As seen from the overall caries prevalence was slightly higher from 78% to 83%.The conclusion was not scientifically sound based on the results presented in this format.

Response 6:  line 154

 In this research, in Ulaanbaatar, the overall caries prevalence were slightly higher from 78.0%(2017) to 82.6%(2019). However, comparing 2017 and 2019, the proportion of deft > 4 decreased as follows, 61.5% to 46.0% for 3 years old, 84.1% to 60.0% for 4 years old, and 84.2% to 63.6% for 5 years old.

Point 7. Several grammatical errors were detected. Substantive editing is required before publication.

Response 7:

 Authers ask English editing service by MDPI.

Reviewer 2 Report

  1. the whole amuscript should be arranged and divided into 2 groups: children0-3years and children 3-5 years.
  2. the introduction part should be expanded with explanation of not only ECC caries, but sugar intake, breast feeding, oral hygiene, caries in general etc 
  3. important to have analysis about high and low socioeconomic status of family becaus it is known for main reason in caries prevalence
  4. name and initials of authors who made clinical examinations
  5. explain where clinical examinations were done and under which standards?
  6. all materials and results part should be rewritten and divided into two groups; 0-3y and 3-5y and whole statistics should be based on those two groups. It is hard to explain how the children age 0-1y have caries, and how did they brush teeth in that age 
  7. in some figures, change Deft inti deft
  8. table 2, there should be two columns  by age group, 0-3y and 3-5y
  9. contribution of all authors, according to IJERPH instructions
  10. discussion part should be rewritten and expanded;compare and analyse results of ECC of other authors, e.g. in Asia, surrounding countries. Analyse the reasouns for high/low ECC caries, caries prevalence in children, influence of Socio economic status, etc
  11. add more, new and important references. Now is only 8 references for this study?
  12. was this cross sectional observational study or cohort prospective study becaus eit is unclear when authors state that e.g. "The deft in October 2017 and in October 2019 is shown in Fig.4. At age 3, deft of 2019 was significantly lower than 2017 (p<0.01) and at age 4, deft of 2019 was lower tendency than 2017 (p=0.085).

Author Response

Point 1: the whole m anuscript should be arranged and divided into 2 groups: children 0-3 yearsand children 3-5 years

Response 1: 

0-1 year old (0-23 months) children examined at the District Health Center and 2-5 years old ( 24-71 months) children examined at the public No.129 kindergarten. Authors do not think to divide children into 2 groups.

Point 2: the introduction part should be expanded with explanation of not only ECC caries, but sugar intake, breast feeding, oral hygiene, caries in general etc

Response 2: Introduction part was expanded. line 32

 Dental caries is highly prevalent worldwide, the significant public health problem and reduce the quality of life [1,2].The majority of caries lesions are concentrated in few, often disadvantaged social groups. A systematic review of socioeconomic inequality and caries showed that low socioeconomic position is associated with a higher risk of having caries. The association between low educational background and having DMFT/deft >0 was significantly increased in highly developed countries [3]. The association between breastfeeding and dental caries is complicated. Breastfeeding during the day beyond the age of 12 months was not associated with caries, but infants who were breastfed at night > 2 times associated with caries [4]. Most of the children with rampant or nursing caries had an unbalanced diet with high sugar content. Infant of breast-fed for less than 6 months or longer than 12 months showed rampant caries. Their meels were unusually sweetened in infancy [5].

Point 3: important to have analysis about high and low socioeconomic status of family because it is known for main reason in caries prevalence

Response 3: line 149-150

  Socioeconomic status of the recipient in this study was the middle-class families. Higher parental educational attainment at 3 years old in 2019, deft was lowere than 2017. Glogal evidence suggests that higher educational attainment is associated with better health outcomes.

Point 4:name and initials of authors who made clinical examination

Response 4: line 66

Dr. Mizuho Nishino (MN) and Dr. Saki Hirao (SH)

line 66: 2 Japanese dentists (MN and SH)

Point 5: explain where clinacal examination were done and under which standard?

Response 5: line 58-60, line 77

line 58-60: Taget area was urban area in Mongolia: 18 Khoroo, Bayansurkh district, Ulaanbaatar city. Target groups were the dentists, the parents and the children. The chief of the District Health Center called 0-1 year old (0-23 months) children to come to the District Health Center. The chief of public No.129 kindergarten informed to parents of 2-5 years old (24-71 months) children.  

line 77: The knee-to-knee oral examination with "Oral Health Assessment Form for Children in Mongolia based on World Health Organization"was performed by two Japanese dentists (MN and SH) using .........

Point 6:all materials and results part should be rewritten and divided into two groups: 0-3y and 3-5y and whole statistics should be based on those two groups. It is hard to explain how the children age 0-1y have caries, and how did they brush teeth in that age

Response 6:

As stated at Response 1, authors do not think to divide children into 2 groups.

line 141: Japanese dentists educated the Mongolian dentists and parents/gurdians of children how to brush the infant's teeth like as illustrated in Figure 1, educational sheet.

Point 7: in some figures, change Deft into deft

Response 7: Deft was changed to deft.

Point 8:table 2, there should be two columns by age group, 0-3y and 3-5y

Response 8: Authors do not think to divide children into 2 grops.

Point 9: contribution of all authors, according to IJERPH instruction

Response 9: Author Contributions are shown between Authors and Coverletter. And at line 177.

Point 10: discussion part should be rewritten and expanded, compare and analyse results of ECC of other authors,e.g. in Asia, surrounding countries. Analyse the reasons for high/low ECC caries, caries prevalence in children, influence of Socio economic status, etc

Response 10: Discussion part was rewritten and expanded.

Point 11: add more, new and impotant references. Now is only 8 references for this study?

Response 11: Introduction and discussion parts were rewritten and expanded. Then referances 1,2,3,4,5,14,15,16 were added.

Point 12: was this cross sectional observational study or cohort prospective study because it is unclear when authors state that e.g." The deft in October 2017 and in October 2019 is shown in Fig.4. At age 3, deft of 2019 was significantly lower than 2017 (p<0.01) and at age 4, deft of 2019 was lower tendency than 2017 (p=0.085).

Response 12: Discussion part was rewritten.

 This study was repeated cross sectional observational study to examine the effect of this project. In this research, in Ulaanbaatar, the overall caries prevalence was slightly higher from 78.0%(20017) to 82.6%(2019). However, comparing 2017 and 2019, the proportion of deft >4 decreased as follows, 61.5% to 46.0% for 3 years old, 84.1% to 60.0% for 4 years old, and 84.2% to 63.6% for 5 years old.

Reviewer 3 Report

English must be improved as some sentences do not even have an structured meaning and makes it difficult to understand.

It is not clear if the children visited at different stages were the same ones or not. It is not clear that the children visited were the children of parents receiving instruction and education. There is not difference in data regarding caries prevalence among children using fluoride and non-fluoride toothpaste.

Conclusions should be more detailed.

Author Response

Point 1: English must be improved as some sentences do not even have an structured meaning and makes it diddicult to understad.

Response 1: Authors ask English editing service by MDPI.

Point 2: It is not clear if the children visited at different stages were the same ones or not.

Response 2: line 140 addition

 In this study, the number of children was all children who came for oral examination. Researcher planed follow-up of all children,but in Mongolia almost all parents work and it was difficult to come every time for oral examination. Definition of the project target has not changed but its contents have changed and each individual contains different people. This study was repeated cross sectional observational study to examine the effect of this project.

Point 3: It is not clear that the children visited were the children of parents receiving instruction and education.

Response 3:

 Every time, education to parents, taking questinnaires and oral examination were done.

Point 4: There is not difference in data regarding caries prevalence among children using fluoride and non-fluoride toothpaste.

Response 4:

 In this study, the difference of caries prevalence between children using fluoride and non-fluoride tooth pastes was not analysed. Because in Mongolia, fluoride concentration is not clear, may be very low. In questionnaire, researcher intended to know parent's concern to fluoride containing toothpaste.

Point 5: Conclusion should be more detailed.

Response 5: To the conclusion part,  study method was added as follows:

 In urban area in Mongolia, six times between October 2017 and October 2019, the education for caries prevention, taking questionnaire for daily oral health behavior and oral examination for parents and children aged 0-5 years old were done. Total participant was 2,223 and the repeated cross sectional observational study was done.

Round 2

Reviewer 1 Report

I am satisfied with the major revision.

Author Response

  1. Referances were corrected as IJERPH style.
  2. In conclusion and abstract, the sentence that target population of parents were middle socio-economic famillies was inserted.

Reviewer 2 Report

references must be corrected as IJREPH style: Journal Articles:
1. Author 1, A.B.; Author 2, C.D. Title of the article. Abbreviated Journal Name YearVolume, page range.

e.g. reference No.17, No.4, No.9...

2. insert in conclusions and abstract that target population of parents were middle socio-economic families. 

3. because this study used "field dental equipment and analysis", there is some limitations in contrast to epidemiology studies

Author Response

  1. References were corrected as IJERPH style.
  2. In conclusion and abstract, the sentence that target population of parents were middle socio-economic families was inserted.